# Win Statistics in Observational Cancer Research: Integrating Clinical and Quality-of-Life Outcomes

**DOI:** 10.3390/jcm13113272

**Published:** 2024-05-31

**Authors:** Maria Vittoria Chiaruttini, Giulia Lorenzoni, Gaya Spolverato, Dario Gregori

**Affiliations:** 1Unit of Biostatistics, Epidemiology and Public Health, Department of Cardiac, Thoracic, and Vascular Sciences and Public Health, University of Padova, 35128 Padova, Italy; mariavittoria.chiaruttini@studenti.unipd.it (M.V.C.); giulia.lorenzoni@unipd.it (G.L.); 2Department of Surgical Oncological and Gastrointestinal Science, University of Padova, 35128 Padova, Italy; gaya.spolverato@unipd.it

**Keywords:** chronic diseases, cancer, etiology and risk factors of diseases, win statistics, propensity score, observational studies, quality of life

## Abstract

**Background**: Quality-of-life metrics are increasingly important for oncological patients alongside traditional endpoints like mortality and disease progression. Statistical tools such as Win Ratio, Win Odds, and Net Benefit prioritize clinically significant outcomes using composite endpoints. In randomized trials, Win Statistics provide fair comparisons between treatment and control groups. However, their use in observational studies is complicated by confounding variables. Propensity score (PS) matching mitigates confounding variables but may reduce the sample size, affecting the power of win statistics analyses. Alternatively, PS matching can stratify samples, preserving the sample size. This study aims to assess the long-term impact of these methods on decision making, particularly in colorectal cancer patients. **Methods**: A motivating example involves a cohort of patients from the ReSARCh observational study (2016–2021) with locally advanced adenocarcinoma of the rectum, situated up to 12 cm from the anal verge. These patients underwent either a watch-and-wait approach (WW) or trans-anal local excision (LE). Win statistics compared the effects of WW and LE on a composite outcome (overall survival, recurrence, presence of ostomy, and rectum excision). For matched win statistics, we used robust inference techniques proposed by Matsouaka et al. (2022), and for stratified win statistics, we applied the method proposed by Dong et al. (2018). A simulation study assessed the coverage probability of matched and stratified win statistics in balanced and unbalanced groups, calculating how often the confidence intervals included the true values of WR, NB, and WO across 1000 simulations. **Results**: The results suggest a better efficacy of the LE approach when considering efficacy outcomes alone (WR: 0.47 (0.01 to 1.14); NB: −0.16 (−0.34 to 0.02); and WO: 0.73 (0.5 to 1.05)). However, when QoL outcomes are included in the analyses, the estimates are closer to 1 (WR: 0.87 (0.06 to 2.06); WO: 0.93 (0.61 to 1.4)) and to 0 (NB: −0.04 (−0.25 to 0.17)), indicating a negative impact of the treatment effect of LE regarding the presence of ostomy and the excision of the rectum. Moreover, based on the simulation study, our findings underscore the superior performance of matched compared to stratified win statistics in terms of coverage probability (matched WR: 97% vs. stratified WR: 33.3% in a high-imbalance setting; matched WR: 98% vs. stratified WR: 34.4% in a medium-imbalance setting; and matched WR: 99.2% vs. stratified WR: 37.4% in a low-imbalance setting). **Conclusions**: In conclusion, our study sheds light on the interpretation of the results of win statistics in terms of statistical significance, providing insights into the application of pairwise comparison in observational settings, promoting its use to improve outcomes for cancer patients.

## 1. Introduction

Composite endpoints are widely used in clinical studies. They consist in pooling multiple outcomes together. Reasons for pooling are based on clinical reasoning [1] and statistical convenience. Regarding the latter, pooling helps overcome the limitations related to low-prevalence outcomes, resulting in smaller sample sizes and a lower study duration [2].

In cancer research, the most commonly used composite endpoint is progression-free survival, including both death and cancer progression. However, the composite endpoint offers the opportunity to include softer endpoints such as quality of life (QoL) and patient preferences, which are topics of great interest [3,4]. This issue is particularly relevant in certain types of diseases. For instance, colorectal cancer operated with curative intent has a generally low mortality rate in follow-up, but its treatment may require extensive procedures, like the creation of a stoma, which can potentially significantly impact the patient’s QoL, especially in Mediterranean cultures [5]. The composite endpoint can allow for the integration of these components.

However, despite the advantages that have made composite endpoints so popular in clinical studies, clinical epidemiologists are skeptical about their use. Using composite endpoints presents with several limitations, and the main one is the lack of prioritization of clinical outcomes, i.e., outcomes are pooled all together and only those occurring first are collected without considering what are more relevant for the health of the patient (e.g., hard complex outcomes such as death are placed on the same level as softer outcomes) [6].

Win statistics such as Win Ratio (WR), Win Odds (WO), and Net Benefit (NB) [7] provide a solution for prioritizing outcomes, maintaining the statistical advantages of the composite endpoints.

The most common win statistics approach is the unmatched pairs one, where each participant in the treatment group is compared with all patients in the control group (“all against all” approach, where total pairs = patients in the treatment group × patients in the control group). For each pair it is evaluated if the treated patient is a winner or loser within each component of the composite in a hierarchical order.

However, since the win statistics have been initially crafted for randomized controlled trials (RCT) [8,9,10,11,12,13,14,15,16,17,18,19,20,21], their transition to an observational setting introduces challenges related to confounding variables, necessitating a tailored approach for effective implementation. In fact, unlike RCT, where participant recruitment is carefully managed to ensure homogeneity among arms [22], observational studies are specifically designed to include the inherent variability of real-world scenarios and play a pivotal role in unraveling real-world complexities, offering insights that randomized controlled trials may not capture [23]. Consequently, in the observational setting, the conventional “all against all” pairing approach lacks the assurance of the control of confounding variables [24].

An alternative win statistics approach made on matched pairs has also been suggested, aiming to form pairs with similar baseline risk profiles. This approach could be considered most suitable within observational settings where the problem of potential confounding related to the non-random allocation of the patients in the treatment groups is relevant and cannot be adequately accounted for with the unmatched pair approach.

In this regard, the propensity score (PS) could provide an efficient tool for precise one-to-one matching between treated and untreated individuals, where pairs are established to ensure balanced comparisons across arms concerning confounding factors [25,26].

The PS technique, while potentially reducing bias, is challenging due to the difficulty in prospectively identifying all relevant adjustment variables. Furthermore, the propensity score matching often leads to a significant reduction in sample size, which becomes more pronounced as the imbalance between groups increases [27]. While this may not pose a statistical challenge, thanks to applying a robust inference [20], it presents an interpretative dilemma. By trimming the tails of the distribution, the act of matching leads to a loss of variability, automatically forfeiting the inherent advantage of observational studies, which is their real-world/pragmatic approach.

Another strategy is the stratified approach. This involves pre-identifying key risk factors for outcomes and dividing participants into strata based on these factors before conducting pairwise comparisons. The stratum-specific win statistics are then pooled to estimate the study’s overall WR, WO, or NB.

This paper aims to investigate the statistical treatment of composite endpoints within colorectal surgery using win statistics in a non-randomized setting and applying the propensity score both in a classical matching approach and by stratifying the sample into groups with a similar probability of receiving the treatment.

In doing this, a real-world case study in the colorectal cancer field has been designed. Moreover, we proposed a simulation study to illustrate how confounding factors impact the estimation of win statistics when the “all against all” approach is used in observational settings and to evaluate the ability of matched and stratified win statistics to correct biased estimations and produce reliable statistics.

## 2. Materials and Methods

### 2.1. Real Case Study in Colorectal Cancer Research: The ReSARCh Observational Study

The real-case application focused on a cohort of patients enrolled in the ReSARCh observational study [28] between 2016 and 2021. The study included patients with locally advanced, histologically confirmed adenocarcinoma of the rectum, situated up to 12 cm from the anal verge. These patients underwent a rectum-sparing approach via either a watch-and-wait approach (WW) or trans-anal local excision (LE).

Although various studies presented the results of LE and WW in treating patients with locally advanced rectal cancer, reliable data comparing the outcomes of these two approaches are still lacking [29]. In part, the reason WW and LE are difficult to compare is the need to consider different outcomes, including survival, relapses, and QoL variables [30,31]. In this framework, the win statistics could easily consider both classical efficacy and QoL outcomes, providing a comprehensive evaluation of treatment benefits and drawbacks. Thus, the outcomes that are included, ranked by importance, are (i) the overall survival (OS), (ii) the distant recurrence (DR), (iii) the local recurrence (LR), (iv) the presence of ostomy, and (v) the excision of the rectum.

The OS, DR, and LR have been considered as standard efficacy outcomes, while the presence of ostomy and the excision of the rectum have been considered as QoL outcomes.

Due to the observational nature of the case study, the PS was calculated by a generalized linear model using Sex, Age, Body Mass Index (BMI), Tumor distance from anal verge, and Tumor Stage (i.e., ycT = 0, ycT ≥ 1) at restaging after neoadjuvant chemoradiotherapy (nCRT) as predictors.

Finally, the matched and stratified WR, NB, and WO were calculated using both the efficacy outcomes only and the efficacy plus the QoL outcomes together.

The ReSARCh study was approved by the Institutional Review Board of the Hospital of Padua, Italy. This study was conducted in accordance with the tenets of the Declaration of Helsinki and is registered at clinicaltrials.gov under protocol number NCT02710812. Informed consent was not required for this study because the data were obtained from a de-identified patient database.

### 2.2. Simulation Study

The simulation study aimed to illustrate how confounding factors impact the estimation of win statistics in observational setting, offering a broader understanding of the behavior of the matched and stratified win statistics after the PS approach.

#### 2.2.1. Randomized Controlled Trial (RCT) Setting

In order to obtain unbiased estimations (free from confounding factors) of the WR, WO, and NB, a simulated RCT dataset balanced among arms was created.

The simulated dataset contained 200 patients, randomized 1:1 to the treatment and control groups.

Two time-to-event endpoints (E^s^ with s = 1, 2) and three covariates (X1, X2, and X3) plus the treatment variable were simulated. The censoring times were simulated as two exponential variables (C^s^ with s = 1, 2) for both treated and control patients, as follows (where the subscript *i* = 1,…,N represents the subject index):$$C_{i}^{s}=S_{i}^{-1}(t)=\frac{-log(t)}{\lambda exp(X_{i}^{T}\beta )}.$$

The survival times, T^s^ with s = 1, 2, were simulated using the Weibull regression model, as follows, where *g* = *C*, *T* represents the control and treatment groups index:$$T_{i}^{s}=S_{i}^{-1}(t)={\left(\frac{-log(t)}{\lambda_{g,s}exp(X_{i}^{T}\beta )}\right)}^{\frac{1}{\gamma_{g,s}}}.$$

The true effect of the covariates (i.e., log hazard ratios) have been set at discretion, equal to −1.9 for the treatment group, 0.8 for the X1 covariate, −0.3 for the X2 covariate, and 1.7 for the X3 covariate. Thanks to randomization, the treated and control groups resulted in a balance for the baseline covariates.

The win statistic estimations on the RCT dataset have been considered “true” estimations of the treatment effect on the composite outcome.

#### 2.2.2. Observational Setting

Subsequently, three observational datasets were generated to assess the ability of the matched and stratified win statistics to correct the estimates when applied in the presence of confounders. Each dataset contains 200 patients with a 1:1 allocation and corresponds to a diverse scenario of high, medium, and low levels of imbalance. The covariate effects (i.e., log hazard ratios for the treatment, X1, X2, and X3 covariates) have been kept equal to the RCT setting.

Finally, the matched and stratified win statistics were computed to evaluate their properties in terms of coverage probability. The coverage probability was calculated as the frequency (across 1000 simulations) with which the computed confidence intervals of matched and stratified win statistics encompassed the true values of WR, NB, and WO defined in the RCT setting. The PS was calculated by an additive generalized linear model using X1, X2, and X3 as predictors. All the analyses have been performed using R for Windows 4.3.3 [32].

The data presented in this study are available on request from the corresponding author.

### 2.3. Win Statistics

Win statistics such as WR, WO, and NB [6,7] have been proposed as a solution for prioritizing outcomes. All these methods compare subjects in the treatment group with the subjects in the control group, and each comparison has three possible results: the subject in the treatment group wins, the control patient wins, or the two patients are tied. For example, the treatment patient would “win” if the control patient died earlier from a cancer disease. Every comparison begins by considering the highest-priority outcome, and only when this initial comparison results in a tie does the analysis factor in the following most crucial outcome. If each comparison, based on the individual outcomes, ends in a tie, then the overall result for that pair is also considered a tie. Consequently, lower-priority outcomes do not obscure the significance of more essential outcomes solely because they occur earlier in the analysis.

Let *πt* be the probability associated with the treatment patient winning, *πc* be the probability associated with the control patient winning, and *πtie* represent the probability of a tie (*πt* + *πc* + *πtie* = 1.0). The subscripts t and c denote the treatment and control groups, respectively. The definitions of *WR*, *WO*, and *NB* are as follows:$$WR=\frac{\pi t}{\pi c}\;WO=\frac{\left(\pi t+0.5\pi tie\right)}{\left(\pi c+0.5\pi tie\right)}\;NB=\pi t-\pi c.$$

### 2.4. Win Statistics in Randomized Studies

#### “All against all” WR, WO, and NB

In the context of an RCT, the “all against all” analysis is used, and *NtNc* (*Nt* = number of patients in the treatment group; *Nc* = number of patients assigned to the control group) pairwise comparisons are generated. Thus, the corresponding estimates of the win probabilities *πt*, *πc*, and *πtie* are the win proportions (where nt and nc are the number of wins for the treatment group and control group, respectively [7]):$$Pt=\frac{nt}{NtNc}\;Pc=\frac{nc}{NtNc}\;Ptie=1-Pt-Pc.$$

For the construction of the confidence interval of the win statistics estimates, the normal approximation can be applied, and the variances of *log*(*WR*), log(*WO*), and *NB* under the null hypothesis can be estimated by:$${\hat{\sigma }}^{2}log(WR)=\frac{\left({\hat{\sigma }}^{2}t-2{\hat{\sigma }}^{2}tc+{\hat{\sigma }}^{2}c\right)}{{\left(\frac{nt+nc}{2}\right)}^{2}}\;{\hat{\sigma }}^{2}log(WO)=\frac{\left({\hat{\sigma }}^{2}t-2{\hat{\sigma }}^{2}tc+{\hat{\sigma }}^{2}c\right)}{{\left(\frac{NtNc}{2}\right)}^{2}}\;{\hat{\sigma }}^{2}NB=\frac{\left({\hat{\sigma }}^{2}t-2{\hat{\sigma }}^{2}tc+{\hat{\sigma }}^{2}c\right)}{{\left(NtNc\right)}^{2}}$$
where ${\hat{\sigma }}^{2}t,\;{\hat{\sigma }}^{2}c$ are the estimated variances for nt and nc, respectively, and ${\hat{\sigma }}^{2}tc$ is their estimated covariance. Details are given by Dong et al. [14] and Bebu and Lachin [33].

### 2.5. Win Statistics in Non-Randomized Studies

In comparing two patients in non-randomized studies, it is advisable to consider the underlying risk of outcomes. In fact, it would be an unfair comparison if one was at a higher risk of developing the outcomes compared to another one with a lower baseline risk [6]. In addition, patients within the treatment and control groups are likely to be dependent, resulting in an overestimation of precision and thus a biased inference [34]. In such contexts, methods like PS are applied to reconstruct a quasi-randomized scenario [20]. Mainly, PS can be used in a classical matching approach or to stratify the patients into groups with a similar probability of receiving the treatment.

#### 2.5.1. Matched WR, WO, and NB

Propensity score matching involves the creation of matched pairs consisting of individuals who have received treatment and those who have not, and who exhibit similar propensity score values [25,26]. This approach enables the estimation of the Average Treatment effect on the Treated (ATT) [35]. In the matched framework, the estimates of the win probabilities *πt* and *πc* are the win proportions (where the denominator is the number of *N* pairs of patients):$$Pt=\frac{nt}{N}\;Pc=\frac{nc}{N}.$$

This classical approach, largely referred in the initial work of Pocock [6], presents some limitations [20], in particular for what concerns the small sample properties of the Wald estimator. In this paper, a modified approach for robust inference in the case of matched NB, WR, and WO—recently proposed by Matsouaka et al. in 2022 [20]—has been used.

#### 2.5.2. Stratified WR, WO, and NB

Stratification based on the PS involves categorizing subjects into distinct, non-overlapping subsets according to their estimated propensity scores. Within each propensity score stratum, treated and untreated subjects exhibit similar propensity score values. The win statistics are computed using the “all against all” approach within each stratum and then combined by the Dong method [27,36].

The stratified win proportion *Pt* and *Pc* for the treatment and control groups are calculated as follows (where *M* represents the total number of strata, ${Nt}^{m}{\;\mathrm{and}\;Nc}^{m}$ the total number of treatment and control patients in each stratum, ${nt}^{m}$ and ${nc}^{m}$ the number of wins for the treatment group and the control group in each stratum, and $w^{m}$ the weight for the *m^th^* stratum):$$Pt=\frac{\sum_{m=1}^{M}w^{m}nt^{m}}{\sum_{m=1}^{M}w^{m}Nt^{m}Nc^{m}}\;Pc=\frac{\sum_{m=1}^{M}w^{m}nc^{m}}{\sum_{m=1}^{M}w^{m}Nt^{m}Nc^{m}}$$

In this paper we weighted the wins with the reciprocal of the stratum size following the Mantel–Haenszel-type stratified analysis as described in Dong et al. (2018) [36].

## 3. Results

### 3.1. Real Case Study

The ReSARCh study’s analysis included 190 participants, 117 of whom underwent LE and 73 of whom received the WW approach. Table 1 presents the baseline characteristics and outcome distributions in the two groups.

After matching the data, each group included only 51 patients. Eighty-eight (47%) patients were excluded from the original sample, mainly those with ycT ≥ 1 (70 (60.9%) before matching vs. 29 (28.4%) after matching, leading to a loss of information regarding patients with a not-complete response after nCRT. The covariate balance after matching is reported in Figure 1.

For what concerns PS stratification, the first stratum comprised 36 patients (33 LE, 3 WW), the second had 35 patients (26 LE, 9 WW), the third had 35 patients (22 LE, 13 WW), the fourth included also 35 patients each (17 LE, 18 WW), and the fifth had 36 patients (13 LE and 23 WW).

Table 2 presents the standardized mean difference (SMD) for each covariate of interest in the ReSARCh study, using the “all against all” approach before and after adjustment by matching and stratification.

As expected, the unadjusted SMD has values greater than 0.1, except for Sex. However, even the stratified approach failed to achieve a balance of covariates between treatment groups: although there is a reduction in SMD on average across strata, it is not consistent for stratum-specific values. The approach that resulted in the best balance of the covariates was the matching one, even though it resulted in a considerable loss of patients.

The results of the win statistics estimated in the matched and stratified samples, calculated first using efficacy outcomes only and then including also QoL endpoints, are reported in Table 3.

The results suggest a better efficacy of the LE approach when considering efficacy outcomes alone. However, when QoL outcomes are included in the analyses, the estimates are closer to 1 (for WR/WO) and to 0 (for NB), indicating a negative impact of the treatment effect of LE regarding the presence of ostomy and the excision of the rectum. Furthermore, while the results are consistent, there is a significant difference in the width of the 95% confidence intervals between matched and stratified estimates, with stratified statistics presenting narrower CIs. This leads to significant results for stratified win statistics considering efficacy outcomes only, as the upper bound results were below unity for both WR and WO, and below null for the NB estimate.

### 3.2. Simulation Study

Appendix A graphically depict the survival probabilities over time for E1 and E2 events within the RCT framework through Kaplan–Meier curves when the treatment group has a better survival probability than the control group for both E1 (median survival: treated = 16.7 vs. control = 6.95) and E2 (median survival: treated = 14.4 vs. control = 5.54).

Utilizing the “all against all” methodology within the RCT-simulated dataset, the “true” WR, NB, and WO were calculated to be 3.76 (95%CI: 2.4 to 5.9), 0.46 (95%CI: 0.28 to 0.63), and 2.67 (95%CI: 1.88 to 3.8), respectively.

Regarding the simulated observational datasets, Appendix A outlines the baseline characteristics of synthetic participants when the imbalance has been included among arms. Moreover, the Kaplan–Meier curves presented in Appendix A present the survival distribution within these datasets, showing less pronounced differences between the interest groups in the survival probabilities at follow-up despite the fact that the data were simulated by assuming the effect of the treatment was non-zero. This attenuation is attributed to a disproportionately higher number of patients with adverse prognostic factors in the treatment arm relative to the control group.

Table 4 presents the SMDs both unadjusted and after PS matching and PS stratification (average across strata) at the three different degrees of imbalance.

The matched datasets demonstrate a good balance, with SMDs around or below 0.1. In contrast, the datasets do not achieve a perfect balance after PS stratification at each level of imbalance, which is consistent with the results obtained within the application of matching and stratification techniques within the real dataset.

Table 5 provides the coverage probability based on 1000 simulations, reflecting the frequency with which the confidence intervals of unadjusted (“all against all” approach), matched, and stratified win statistics calculated in the observational setting encompass the true values of WR (3.76), NB (0.46), and WO (2.67).

This metric serves as an indicator of the robustness and reliability of the interval estimates in capturing the actual treatment effects under study.

Even if the PS matching has led to notable reductions in the sample size, i.e., 26%, 38%, and 50% for low-, medium-, and high-imbalance scenarios, respectively, matched WR and WO exhibited superior coverage probabilities, consistently surpassing the 95% threshold, irrespective of the level of imbalance present in the data. In contrast, among the stratified win statistics, the NB results as the most reliable estimation, reporting a coverage probability close to 80% in scenarios with high imbalance and approximately 95% when the imbalance is moderate to low.

## 4. Discussion

### 4.1. Main Results

The present study aimed to explore the most suitable approach for delving with composite endpoints using win statistics within the observational study setting.

This study applied a simulation approach to a real case study. Overall, the results show that the matching approach results in a better covariate balance, albeit at the cost of a greater loss of patients. Furthermore, the simulation study showed that the matching generally resulted in higher coverage probabilities within the simulation setting, regardless of the level of imbalance.

Current opinions in research on colorectal cancer therapy underscore the potential of WR statistics to provide a more nuanced understanding of the balance between treatment efficacy and QoL [37]. While there are inherent technical challenges in employing win statistics, they can significantly enhance the comprehensive assessment of the advantages and disadvantages of treatments, particularly for patients with a longer expected lifespan, as is often the case with colorectal cancer patients [38].

Comparative analysis of the ReSARCh cohort, employing both matched and stratified win statistics, indicates that the WW approach might result in less effective outcomes compared to the LE strategy. However, this difference is less marked when considering QoL metrics, implying that the impact of the LE procedure on QoL might offset some of its advantages. It should be noted that the matched win statistics are preferable for the ReSARCh data analysis due to the improved balance achieved through matching rather than stratification. On the other hand, the matching process, despite resulting in higher coverage probabilities, has resulted in a loss of information concerning patients with an incomplete response after nCRT. For this reason, caution should be exercised in interpreting inferential results, and generalizations should be limited to the characteristics of the paired population.

### 4.2. Matched Win Statistics and Related Limitations

From the statistical point of view, while PS matching can lead to notable reductions in sample size, Matsouaka et al. (2022) [20] have suggested a robust inference for win statistics. This method compensates for the smaller sample sizes by widening confidence intervals. Indeed, when a study’s sample size is limited, the statistical power to detect true differences between treatments are reduced, potentially leading to less reliable conclusions [39]. To address the impact of small sample sizes, researchers often employ statistical techniques that adjust for the reduced power. One such method involves using wider confidence intervals, reflective of the greater uncertainty inherent in studies with fewer participants [40]. This adjustment ensures that the confidence intervals maintain the appropriate level of coverage probability, even when the sample size is small [41]. Therefore, the inferential interpretation of win statistics must consider that the width of the confidence intervals can lead to unbiased but non-significant results, especially in the context of cohorts characterized by high trimming during PS matching.

### 4.3. Stratified Win Statistics and Related Limitations

On the other hand, a stratified approach could lead to more precise estimates with shorter confidence intervals, as it maintains the original study sample size. However, one significant limitation identified during the analyses is the number of strata when applying the PS. Our simulations revealed that creating five strata based on PS, as recommended by Cochran (1968) [42] and Rosenbaum and Rubin (1984) [43], who showed that such stratification could reduce bias from measured confounders by roughly 90%, can be challenging in the presence of a high imbalance due to insufficient patients in some strata. Moreover, when the size of each stratum is limited due to a high imbalance and/or a small sample, achieving balance within each stratum becomes challenging. This complicates the reconstruction of a quasi-randomized scenario, as demonstrated by our simulations and real-data analysis.

However, researchers handling extensive datasets encompassing numerous patients and predictors may enhance stratification success through optimization analyses and calibration of the PS model. In fact, identifying the optimal model for predicting treatment probability before computing stratified win statistics becomes crucial for the purpose of win statistics adjustment.

### 4.4. Final Remarks

In our simulation setting and real case study, the matching approach ensures the optimal balancing of confounding variables, leading to matched win statistics with a higher coverage probability, as demonstrated by our results. However, the matching also leads to a greater loss of patients, necessitating cautious interpretation of win statistics results.

In fact, applying robust inference techniques to face the reduced sample size may render the results non-significant with large confidence intervals. Hence, when employing matched win statistics, it is imperative to evaluate the adequacy of the sample size to ensure sufficient statistical power for rejecting the null hypothesis in the presence of a true underlying treatment effect [44].

On the other hand, the stratified approach could be a viable alternative to offset the sample size reduction resulting from matching. However, achieving covariate balance within strata to ensure correction for confounding factors in win statistics computation is not straightforward and requires calibrating the PS model.

In conclusion, our study emphasizes practical considerations, providing insights to aid researchers in effectively navigating and addressing the application of win statistics in observational settings.

## Figures and Tables

**Figure 1 jcm-13-03272-f001:**
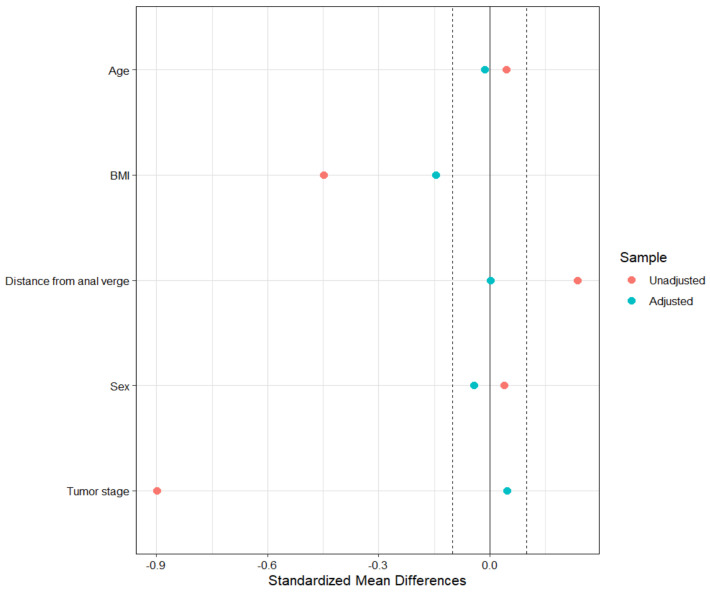
Balance plot in the matched sample.

**Table 1 jcm-13-03272-t001:** Baseline characteristics of colorectal cancer patients in the ReSARCh study by local excision (LE) and the watch-and-wait (WW) strategy.

Characteristic	LE, n = 117 ^1^	WW, n = 73 ^1^	*p*-Value ^2^
Sex, Male vs. Female	76 (65%)	48 (65.8%)	0.9
Age, years	65 (60, 72)	65 (58, 72)	0.9
Body Mass Index	26.3 (23.9, 28.9)	24–5 (22.4, 27.4)	0.01
(Missing)	2	5	
Anal verge dist., cm	4.5 (3.00, 6.7)	5.00 (3.00, 7.00)	0.2
(Missing)	2	0	
Tumor Stage, ycT ≥ 1 vs. ycT = 0	70 (60.9%)	17 (23.9%)	<0.001
(Missing)	2	2	

^1^ n (%); median (IQR). ^2^ Pearson’s Chi-squared test; Wilcoxon rank sum test; Fisher’s exact test.

**Table 2 jcm-13-03272-t002:** Standardized mean difference (SMD) unadjusted, after PS matching, and after PS stratification for each covariate of interest in the ReSARCh study.

Covariate	Unadjusted	Matched	Stratified					
			Strata					Mean
			1	2	3	4	5	
Sex	0.02	0.04	0.13	0.37	0.11	0.04	0.17	0.16
Age	0.36	0.01	0.11	0.28	0.24	0.01	0.30	0.19
BMI	0.43	0.14	0.08	0.24	0.24	0.36	0.01	0.18
Anal verge dist.	0.24	0.002	0.17	0.03	0.04	0.17	0.25	0.13
Tumor Stage	0.81	0.04	0.00	0.00	0.29	0.00	0.00	0.06

**Table 3 jcm-13-03272-t003:** Matched and stratified win statistics estimates (95%CI) using ReSARCh data, considering efficacy only and efficacy + QoL outcomes.

Win Statistics	Efficacy Outcomes	Efficacy + QoL Outcomes
Matched		
Win Ratio (95%CI)	0.47 (0.01 to 1.14)	0.87 (0.06 to 2.06)
Net Benefit (95%CI)	−0.16 (−0.34 to 0.02)	−0.04 (−0.25 to 0.17)
Win Odds (95%CI)	0.73 (0.5 to 1.05)	0.93 (0.61 to 1.4)
Stratified		
Win Ratio (95%CI)	0.39 (0.19 to 0.83) *	0.7 (0.36 to 1.32)
Net Benefit (95%CI)	−0.18 (−0.33 to −0.03) *	−0.09 (−0.25 to 0.07)
Win Odds (95%CI)	0.7 (0.52 to 0.94) *	0.84 (0.61 to 1.15)

* Significant estimated effect (with 95% Confidence Interval).

**Table 4 jcm-13-03272-t004:** Standardized mean difference (SMD) unadjusted and after propensity score matching and stratification in the observational setting (simulation study).

Characteristic	Unadjusted	Matched	Stratification
Imbalance: high	n = 200	n = 100	n = 200
X1, dichotomous	0.34	0.08	0.22
X2, continuous	1.32	0.06	0.56
X3, dichotomous	0.30	0.12	0.39
Imbalance: medium	n = 200	n = 124	n = 200
X1, dichotomous	0.26	0.1	0.27
X2, continuous	0.90	0.05	0.31
X3, dichotomous	0.28	0.12	0.29
Imbalance: low	n = 200	n = 148	n = 200
X1, dichotomous	0.2	0.13	0.19
X2, continuous	0.46	0.05	0.21
X3, dichotomous	0.24	0.1	0.12

**Table 5 jcm-13-03272-t005:** Coverage probability calculated across 1000 simulations. It represents the percentage of estimated confidence intervals in the observational setting that contain the true point estimates for WR, NB, and WO (simulation study).

Setting	WR	NB	WO
Imbalance: high			
Unadjusted	0%	0%	0.1%
PS matching	97%	77%	97%
PS stratification	33.2%	80.2%	8.3%
Imbalance: medium			
Unadjusted	1.1%	5.9%	23.6%
PS matching	98%	78%	98%
PS stratification	34.4%	94%	8.9%
Imbalance: low			
Unadjusted	38.1%	59.6%	86.8%
PS matching	99.2%	84.4%	99.2%
PS stratification	37.4%	95.2%	9.6%

## Data Availability

The data presented in this study are available on request from the corresponding author since the data are not publicly available for privacy reasons.

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
