# Peer review of "Win Statistics in Observational Cancer Research: Integrating Clinical and Quality-of-Life Outcomes"

_jcm, 2024, doi:10.3390/jcm13113272_

Round 1

Reviewer 1 Report

Comments and Suggestions for Authors

- Line 132, line 293: should be "as follows" instead of "as follow"

- Line 316: R should be cited here

- Line 345: is there any reason to believe that the smaller sample is less representative, or that the increase in accuracy provided by being able to use the matched sample is not greater than the decrease in power required with the reduction in sample size?

- Line 427: I see you mentioned the decrease in power here - I'm still curious if it is potentially less impactful/accurate to report the results in the same way, just with wider confidence intervals. I'm not sure that there's anything to do about it, especially when you are reporting the limitations as comprehensively as you are. No revisions required here - just food for thought.

Author Response

Thank you very much for taking the time to review this manuscript. Please find the detailed responses and the corresponding revisions/corrections highlighted/in track changes in the re-submitted files.

Please see the attachment for the detailed point-by-point response to your comments.

Reviewer 2 Report

Comments and Suggestions for Authors

Abstract:

The abstract has not been written well.

Background

The statement “Our study investigates the practical implications of employing robust inference techniques for matched win statistics, as proposed by Matsouaka et al. (2022). Additionally, we compare the PS matching and stratification (Dong, 2018) approaches in the win statistics estimation. Moreover, through a real-world application in the colorectal cancer domain and simulation analyses, we elucidate the long-term implications of these statistical approaches on decision-making processes.” Is the goal of the study. Please move this phrase to the end of the background or provide a section as “objectives”.

Material and methods

As in the last paragraph of the “Introduction” explained “a real-world case study in the colorectal cancer field has been designed. Moreover, we proposed a simulation study to illustrate how confounding factors impact the estimation of win statistics when the ‘all against all’ approach is used in observational settings and to evaluate the ability of matched and stratified win statistics to correct biased estimations and produce reliable statistics.”, In the method section you should provide the information regarding “case study in colorectal cancer” and also the “the simulation study”. For instance how these studies have been designed?, how the data have been collected, and Which confounding factors were considered in these studies?

In the material and method section, you should explain how you applied the “win statistics” to your data and how you compared PS matching and stratification approaches in the win statistics estimation. What were your criteria for comparing PS matching and stratification approaches?

Results

In the result section please provide more quantitative statements about the results of applying win statistics using both PS matching and stratification approaches to your data. Then please provide quantitative results of comparing different approaches.

Conclusion

In conclusion, It has been written that “aiding researchers in navigating and optimizing their utilization for enhanced oncological patient outcomes.”; however, the results do not support this statement about optimization. Which optimization algorithm was applied to the method? Please clarify.

Main text:

Material and method

The material and method section is very long and mostly filled with theoretical information of statistical methods. Since it is an original article not just a review article on statistical methods,  I suggest describing the statistical methods on your data (data of real and simulated studies)

In my point of view, it is better to re-write the material and method section as below:

1.       First, provide sufficient information about the real study in colorectal cancer

2.       Provide sufficient information about the simulation study, in this section please also explain the method or software that has been used to simulate the study.

3.       Shortly provide information on applying win statistics to the data of each (real and simulation) study

4.       Provide information about applying PS matching and stratification approaches in win statistics and their differences

5.       Introduce your criteria to compare the approaches.

Comments on the Quality of English Language

line 109, change "does" to "do"

line 164, add "the" underlying risk

line 167, add "an" overestimation

line 209:  change "with respect to" to "Concerning"

line 227 and 288: follow's'

line 260, change has to "have"

line 291: represent's'

line 307: level's'

line 432: 'are' reduced

Author Response

Thank you very much for taking the time to review this manuscript. Please find the detailed responses and the corresponding revisions/corrections highlighted/in track changes in the re-submitted files.

Please see the attachment for the detailed point-by-point response to your comments

Round 2

Reviewer 2 Report

Comments and Suggestions for Authors

After the revision, the manuscript has shown significant improvement.